# Obesity Affects the Microbiota–Gut–Brain Axis and the Regulation Thereof by Endocannabinoids and Related Mediators

**DOI:** 10.3390/ijms21051554

**Published:** 2020-02-25

**Authors:** Nicola Forte, Alba Clara Fernández-Rilo, Letizia Palomba, Vincenzo Di Marzo, Luigia Cristino

**Affiliations:** 1Endocannabinoid Research Group, Institute of Biomolecular Chemistry, National Research Council, Via Campi Flegrei 34, 80078 Pozzuoli (NA), Italy; n.forte@icb.cnr.it (N.F.); a.fernandez@icb.cnr.it (A.C.F.-R.); letizia.palomba@uniurb.it (L.P.); vdimarzo@icb.cnr.it (V.D.M.); 2Department of Biomolecular Sciences, University of Urbino “Carlo Bo”, 61029 Urbino, Italy; 3Canada Excellence Research Chair on the Microbiome-Endocannabinoidome Axis in Metabolic Health, Université Laval, Québec City, QC 61V0AG, Canada

**Keywords:** obesity, SCFAs (Short-chain fatty acids), LPS (lipopolysaccharide), microglia, endocannabinoidome

## Abstract

The hypothalamus regulates energy homeostasis by integrating environmental and internal signals to produce behavioral responses to start or stop eating. Many satiation signals are mediated by microbiota-derived metabolites coming from the gastrointestinal tract and acting also in the brain through a complex bidirectional communication system, the microbiota–gut–brain axis. In recent years, the intestinal microbiota has emerged as a critical regulator of hypothalamic appetite-related neuronal networks. Obesogenic high-fat diets (HFDs) enhance endocannabinoid levels, both in the brain and peripheral tissues. HFDs change the gut microbiota composition by altering the Firmicutes:Bacteroidetes ratio and causing endotoxemia mainly by rising the levels of lipopolysaccharide (LPS), the most potent immunogenic component of Gram-negative bacteria. Endotoxemia induces the collapse of the gut and brain barriers, interleukin 1β (IL1β)- and tumor necrosis factor α (TNFα)-mediated neuroinflammatory responses and gliosis, which alter the appetite-regulatory circuits of the brain mediobasal hypothalamic area delimited by the median eminence. This review summarizes the emerging state-of-the-art evidence on the function of the “expanded endocannabinoid (eCB) system” or endocannabinoidome at the crossroads between intestinal microbiota, gut-brain communication and host metabolism; and highlights the critical role of this intersection in the onset of obesity.

## 1. Introduction

Obesity is characterized by changes in gut microbiota, the onset of low-grade inflammation and increased endocannabinoid tone. Currently, the bulk of research in this field is focused on the gut microbiota since this is where the majority of the bacteria populating the human body are found and nutrition-related changes of taxa composition occur during obesity. In fact, the gut microbiota has arisen as a key factor that regulates host energy homeostasis by participating in the gut–brain crosstalk. It is composed of approximately 3- to 10-fold the amount of cells forming the human body; more than 90% of gut bacteria belong to the three main phyla, namely, Actinobacteria, Bacteroidetes and Firmicutes, which encode for at least 150-fold more singular genes than the human genome [1]. Factors like diet, gender, genetics, geographical location and individual health affect the composition of the gut microbiota and microbiota-produced molecules [2]. Among microbiota-produced molecules, short-chain fatty acids (SCFAs) [3], lipopolysaccharide (LPS) [4], catecholamines [5], 5-hydroxytryptamine (serotonin, 5-HT) [6], glutamate and γ-aminobutyric acid (GABA) [7] are messengers of a bidirectional communication along the gut–brain axis. In recent years, several multifactorial metabolic diseases with increasing incidence like obesity [4,8], type-2 diabetes mellitus [9,10,11], gastrointestinal cancer [12] and inflammatory bowel diseases [13] have been linked with an abnormal microbiome composition, a condition generically named “dysbiosis”, which influences the taxonomical component as well as the metagenomic function, and hence the proteome and metabolome, of the microbial community. Dysbiosis is typically featured by one or more of the following non-mutually exclusive aspects: loss of diversity, loss of commensals and bloom of pathobionts [14]. It is usually characterized by a reduction in the Bacteroidetes/Firmicutes ratio and predominance in taxa causing enteric mucus degradation, thinning of the intestinal barrier and increased gut permeability or gut leakiness [7,9,10,14,15], associated with low-grade chronic inflammation of the intestinal mucosa. Different studies suggest a causal link between obesity and dysbiosis [9]. Gut dysbiosis elicits host inflammatory responses and the ensuing disease status through the production of the potent immunogenic factor lipopolysaccharide (LPS) by the gut Gram-negative bacteria. Seminal studies demonstrated that: (i) infusion of LPS had effects similar to those of a high-fat diet (HFD) at increasing fasting glycaemia, insulinemia and whole-body, hepatic and adipose tissue weight gain; (ii) LPS and endocannabinoid levels are positive correlated [16,17], also because of the downregulation of gene expression of acid amide hydrolase (FAAH, the enzyme that degrades AEA) and *N*-acylethanolamines (NAEs, congeners of AEA) [18]; and (iii) gut dysbiosis and the eCB system are both involved in adipogenesis through a LPS-dependent neuroinflammatory mechanism under obesity-related conditions. In the past several years, our group and others have provided evidence that obesity is associated with increased eCB tone in the brain and plasma, altered expression of cannabinoid receptor 1 (CB1 mRNA) and increased eCB levels in the adipose tissue, liver, muscle and pancreas [19,20,21,22,23,24,25,26,27,28]. Although genetic and pharmacological impairments of peripheral CB1 receptor have been shown to protect against the development of obesity, steatosis and related inflammation [29,30], the molecular link between gut dysbiosis, the eCB system and metabolic disorders associated with obesity remains elusive. Furthermore, it has been suggested that an expanded eCB system or endocannabinoidome (eCBome) encompassing eCB-like mediators, such as the NAEs and other amides, between long-chain fatty acids and neurotransmitters or amino acids and the 2-AG congener monoacylglycerols and the receptors of these lipid mediators, which include transient receptor potential (TRP) channels (such as TRPV1), orphan G-protein-coupled receptors (such as GPR18, GPR55, GPR110 and GPR119), and peroxisome proliferator-activated receptors (PPARs such as PPARα and PPARγ), are part of the picture [31]. Collectively, the present review aims to address studies about the regulatory effect of the LPS-eCB (and eCBome) loop in the gut–brain axis in the context of gut microbiota-mediated inflammatory responses to the host’s high-fat diet. Particular attention will be paid to the role of microorganisms in host energy regulation and LPS- and HFD-induced inflammation as well as the associated LPS-eCB effect on neuronal plasticity via the gut–brain axis during obesity. Targeting microbiota may provide new strategies for therapeutic interventions aimed at preventing or treating obesity and associated metabolic disorders. These strategies include dietary manipulation, such as the use of prebiotics, probiotics or symbiotics, as well as the transplantation of intestinal/fecal microbial communities.

## 2. The Gut Microbiota and Obesity

The microbiome is a community of microbes, Archaea and viruses—and their genes—that collectively accounts for a genome that is 100 times the size of a human so that host genome and the microbiome collectively account for a shared “metagenome” [32]. Bacteria are the predominant members of the gut microbiome, encompassing up to 99% of the genes within the gut [32]. Four main phyla, namely, Bacteroidetes and Firmicutes (representing roughly 90%–99%), Actinobacteria and Proteobacteria [33], are the most abundant in terms of species and strain variability. In the last few years, the idea of an intimate co-evolution between bacteria and humans has been gaining more and more support. Increasing scientific evidence suggests that the gut microbiota and host metabolism influence each other [34,35,36]. Bacteria in the gut help the host in the digestive process by degrading nutrients that are otherwise indigestible, including many plant polysaccharides and complex carbohydrates, which are metabolized to SCFAs such as butyrate, propionate and acetate [37]. Butyrate is used as the primary energy source for colonic epithelial cells, while propionate and acetate are necessary for lipogenesis and gluconeogenesis in the liver [37]. SCFA levels are different between lean and obese mice. Genetic models of obesity, the *ob/ob* mice, have increased levels of butyrate and acetate in the cecal portion of the gut compared to their lean counterparts [37]. These data need to be investigated further in the light of the reported anorexigenic action of acetate in lean mice [3] and the insulin-sensitizing actions of SCFAs in adipocytes and peripheral organs [38], along with their immunomodulatory and anti-inflammatory properties, which participate in the development of the immune system [39]. Germ-free (GF) mice are a critical tool to uncover the causal relationship between the microbiome and disease and to determine the mechanistic basis through which microbes influence the host. Completely devoid of all microorganisms, these mice have contributed to understanding the role of gut microbiota in host fat storage, since these animals have 40% less total fat than wild-type mice and consume on average ~30% more calories [40]. GF mice are protected against obesity after consumption of a Western-style, high-fat, and sugar-rich diet [41], possibly because of the increase in their metabolic efficiency through the upregulation of adenosine monophosphate kinase (AMPK)-mediated activation of PPARs and/or fatty acid oxidation in the skeletal muscle and liver [42]. Fecal transplantation from obese *ob/ob* to GF mice produced an obesogenic effect by enhancing energy harvest of GF mice without modifying their food intake [43], a condition that can be partially prevented by selective transplantation of fecal microbiota enriched with environmental gene tags (EGTs) encoding enzymes that metabolize dietary polysaccharides fibers and producing SCFAs [10,37]. Mechanistic studies revealed that fecal transplantation from obese *ob/ob* to GF mice increased caloric release from polysaccharides and inhibited host genes that counteract energy deposition in adipocytes, including the circulating lipoprotein lipase inhibitor named fasting-induced adipocyte factor (FIAF), which is essential for preventing microbiota-induced deposition of triglycerides in adipocytes [37,42]. Accordingly, GF mice lacking *Fiaf* are vulnerable to diet-induced obesity [42]. Altogether, these findings suggest that gut microbiota is a critical environmental factor that regulates fat storage [42]. Despite the fact that studies linking obesity with microbiota composition have been carried out only in humans, evidence supports that adiposity is transmissible from humans to mice through the gut microbiota, although the underlying mechanisms are still largely unknown [10]. For instance, a study reported that the transfer of lean gut microbiota to obese subjects improved insulin sensitivity in 6 weeks [44], whereas another study showed that transplantation of microbiota from genetically obese humans affected by Prader–Willi syndrome to GF mice produced larger adipocytes and elevation of circulating inflammatory markers [45]. The transplantation of gut microbiota from human twins has shown that GF mice receiving microbiota from the twin obese donor became obese, whereas those receiving microbiota from the twin lean donor remained lean [46].

## 3. The Microbiota–Gut–Brain Axis: Bidirectional Signaling and Routes of Communication

Many homeostatic functions of the gastrointestinal tract and brain are mutually influenced [47]. The contribution of the microbiota to this interaction is becoming increasingly evident to the point of affirming the concept of the microbiota–gut–brain axis. Starting from the beginning of the “Microbial Endocrinology” introduced by Landmark in 1991 [48], the concept of the microbiota–gut–brain axis is gaining ever more interest from investigators, thus helping to increase our understanding of the regulation of homeostatic functions from correlation to causation [47,49]. One of the main interests of microbial endocrinology is based on the shared neurochemical language existing between host and microbes, first demonstrated by Landmark’s studies showing that bacteria respond to host neuroendocrine signaling molecules [48], including 5-HT, GABA, norepinephrine and epinephrine, and that many of these host- and microbial-derived neuroactive signals are also important in host–microbiota interactions at the intestinal interface. It is known that microbiota and brain communicate with each other for the purpose of homeostatic energy regulation of the body using the following chemical signals: (i) LPS, (ii) neurotransmitters, (iii) microbial metabolites (short-chain fatty acids, branched-chain amino acids, peptidoglycans and indole derivatives) and (iv) endocannabinoid-like mediators and *N*-acyl-amides. These molecules move between gut and brain by either being released in the bloodstream or acting on the anatomical pathways of the vagus nerve, enteric nervous system and immune system.

### 3.1. LPS

LPS is formed by three different structural components: O-antigen, core oligosaccharide and lipid A, which is recognized by B cells via cluster of differentiation 14 (CD14) and toll-like receptor 4 (TLR4), thus leading to nuclear factor kappa-light-chain-enhancer of activated B cells (NFkB) activation and the release of pro-inflammatory cytokines, such as tumor necrosis factor α (TNFα) and interleukin 1β (IL1β), and the synthesis of inflammatory oxido/nitrosative enzymes [50,51]. In obesity, along with peripheral and systemic inflammation, gut dysbiosis and LPS-mediated endotoxemia affect brain inflammation by enhancing the passage of circulating inflammatory cytokines across the blood–brain barrier (BBB) [52], stimulating microglia via TLR4 [53] and/or inhibiting vagal afferent neurons [54]. Furthermore, LPS triggers prostaglandin-endoperoxide synthase 2/cyclooxygenase 2 (Ptgs2/COX2) mRNA expression [55] and consequent higher levels/activity of COX2, a pro-inflammatory enzyme that catalyzes the conversion of arachidonic acid to prostanoids. Vasodilation via some prostanoids results in increased migration of lymphocytes and exacerbation of pro-inflammatory TLR4-dependent responses [56], which are associated with the onset of leptin resistance [57]. All these LPS-mediated events are common in the brain of subjects affected by obesity and/or gut dysbiosis [8,58,59], wherein they culminate in several detrimental changes of synaptic plasticity ranging from impairment of memory formation networks in the hippocampus [60,61] to rapid refinement of dendritic spines and lowering of long-term potentiation (LTP) via IL1β-mediated enhancement of GABA tonic current at the GABA receptor [62,63]. Glial cells (microglia and astrocytes) are responsible for the continuous surveillance of the brain in healthy and unhealthy conditions (brain inflammation, damage and progression of several neurological diseases) [64,65]. Following exposure to injury signals, microglia start a rapid reaction accompanied by morphological and functional changes into activated pro-inflammatory (M1) or anti-inflammatory (M2) phenotypes [66]. In the resting state, microglia make tight contacts with synapses and enhance calcium transients in dendritic spines, thereby promoting neuronal synchrony of the neuronal network. On the other hand, LPS-activated microglia impair neuronal activity and promote asynchrony by altering neuronal computation capabilities [67]. The critical role of microglia in synaptic plasticity has been shown in GF mice, which exhibit the immature phenotype and decreased activation of IL1β- and TNFα-mediated inflammatory response, a feature that is reversed by SCFA administration [58]. LPS-activated microglia promote astroglia activation via adenosine triphosphate (ATP) release and consequent increase of glutamatergic synaptic transmission [68], a condition resulting in excitotoxicity and secretion of chemokines, mainly CCL2, from damaged neurons. CCL2 triggers chemotaxis and the migratory activity of activated microglia by stimulating microglial CXCR3 receptors [69]. Accordingly, repetitive LPS-induced activation of microglia causes synaptic stripping of inhibitory terminals in the cortex [70].

### 3.2. SCFAs

The fermentation of dietary fibers by some families of commensal microbes produces SCFAs, namely, butyrate, propionate and acetate [71]. These molecules exert anti-inflammatory properties by counteracting LPS-mediated inflammation [3] and anorexigenic effects, and regulate insulin sensitivity in adipocytes and peripheral organs. SCFAs play these functions usually by binding two G protein-coupled receptors, GPR43 and GPR41 [38,72], to activate the downstream pathways of adenylate cyclase, mitogen-activated protein kinases (MAPK), ion channels and transcription factors. SCFAs can also bypass the cell and nuclear membrane and then block histone deacetylase (HDAC) with consequent epigenetic attenuation of microglial activation and reduction of neuroinflammatory mediators [73]. It was demonstrated that sodium butyrate, the strongest inhibitor of HDAC, improves GABAergic transmission and reduces glutamatergic transmission in opposition to the effect of LPS [74]. SCFAs also enhance memory in contextual fear conditioning [75] and novel object recognition tasks [76]. During brain development, SCFAs play a fundamental function in the development of the BBB [77]. Accordingly, GF mice show increased BBB permeability, a dysfunction rapidly restored by recolonization of the gut using bacterial strains producing mainly butyrate, acetate and propionate [77]. In particular, butyrate is able to induce the maturation of microglia in GF mice via GPR43 [58].

### 3.3. Branched-Chain Amino acids (BCAAs)

The group of branched-chain amino acids is composed of valine, isoleucine and leucine, which are essential amino acids since they cannot be synthesized de novo by mammals and need to be obtained from the diet. They take part directly and indirectly in several biochemical functions, such as protein synthesis, energy production, brain amino acid uptake and immunity in central and peripheral nervous system [78]. Furthermore, BCAAs are key nitrogen donors, involved in intracellular nitrogen shuttling [79]. Despite their vital role in normal physiological functioning, BCAAs can also be toxic and provoke damage in diverse tissues, or can affect neuronal circuits especially in the central nervous system (CNS), when their amounts are disproportionate due to different reasons [80,81]. BCAAs also play a role as regulators of immune-related function, intestinal development and nutrient transporters. Interestingly, gut microbiota produce a higher proportion of BCAAs in comparison with other amino acids, but the real availability of these molecules for the host remains to be elucidated [82]. In particular, the amount of these amino acids in GF mice are altered [83]. Recent studies have shown that supplementation with a BCAA cocktail can favor well-being and even extend the lifespan of mice [84], similar to the benefits obtained with caloric restriction [85,86]. Furthermore, a balanced composition of the gut microbiota in mice could be obtained through BCAA supplementation and caloric restriction [87]. Depending on the BCAA mixture supplementation, the gut microbiota and metabolism will influence, in a different manner, the balance between the Bacteroidetes and Firmicutes phyla [88].

### 3.4. Neurotransmitters

#### 3.4.1. Catecholamines

A variety of bacteria from the human gastrointestinal (GI) tract produce catecholamines with chemical structures identical to those produced by the host. The catecholaminergic system was the first to be found to mediate host–microbe crosstalk. Catecholamines play many roles in the host physiology, from stress-induced fight-or-flight response [89] to influencing gut integrity [90] and affecting host motivational behavior and decision-making [91]. In bacteria, norepinephrine and epinephrine induce a wide range of responses from the promotion of pathogenesis and growth [92] to susceptibility to illness under acute stress caused by norepinephrine-induced bacterial virulence genes, thereby driving infection and mortality [49]. Furthermore, catecholamines can act as siderophores by causing the release of iron from host iron-sequestering proteins and increasing the availability of iron for bacteria, thereby enhancing bacterial growth [93].

#### 3.4.2. GABA

Both host and bacteria have the capacity to convert the amino acid glutamate to GABA [94,95], the major inhibitory neurotransmitter of the host nervous system.

#### 3.4.3. Histamine

Histamine is a biogenic amine that is synthesized from histidine via histidine decarboxylase. This enzymatic pathway is conserved among mammals and certain species of bacteria, and therefore represents an important area of host–microbe communication [5]. In mammals, histamine plays several roles in host physiology, including modulating wakefulness [96].

#### 3.4.4. Indole Derivatives

Direct metabolism of tryptophan (Trp) by microorganisms in the gut includes the direct transformation of this amino acid into several molecules, i.e., indole derivatives that are increasingly recognized as vital in the crosstalk between the host and microbiota. Furthermore, these indole derivatives have an important role in the pathogenesis of metabolic syndrome [97]. Many indole derivatives, such as indole-3-aldehyde (IAld), indole-3-acetic acid (IAA), indole-3-propionic acid (IPA), indole-3-acetaldehyde (IAAld) and indoleacrylic acid, can be ligands for the aryl hydrocarbon receptor (AhR). The AhR signaling pathway is one of the key components in immune response at the intestinal wall, hence it is pivotal for gut homeostasis by contributing to barrier integrity or epithelial renewal and by interacting with diverse immune cell types, such as neutrophils, intraepithelial lymphocytes, innate lymphoid cells, macrophages and Th17 cells [97,98].

#### 3.4.5. Endocannabinoid-Like Molecules (Long-Chain N-acyl-amides)

Recent studies have identified 26 unique commensal bacteria effector genes (Cbegs) that are predicted to encode proteins with diverse catabolic, anabolic and ligand-binding functions and most frequently interact with either glycans or lipids [99]. One such effector gene family (Cbeg12) encodes for the production of N-acyl-3-hydroxypalmitoyl-glycine (commendamide), which was also found in culture from *Bacteroides vulgatus* that harbors a gene highly similar to Cbeg12 [99]. Commendamide resembles long-chain N-acyl-amino acids that function as mammalian signaling molecules through activation of GPCRs or PPARs, such as N-oleoyl-glycine [100], and was found to activate GPCR G2A/GPR132, which has been implicated in autoimmunity and atherosclerosis [99]. More recently, the same authors, by again using bioinformatic approaches, found that some commensal bacteria can produce other N-acyl-amides as well as complex long-chain fatty acids capable of activating the same GPCRs as their host counterparts and, in some cases, GPR119, that belongs to the eCBome [101,102].

### 3.5. Inflammatory Cytokine Signaling

A great concentration of immune cells of the body resides at the luminal–mucosal interface of the GI tract and exchanges immunogenic molecules with the microbiota, thereby directing the immune system to identify potentially harmful pathogens. The microbiota–microglia axis is a very important immunoreactivity interaction occurring both at the intestinal and brain levels. It is mediated by cytokines, chemokines, neuropeptides and neurotransmitters that can pass the blood and lymphatic systems or act on vagal and spinal afferent projections to the brain. Once activated, CNS microglial cells can release a number of cytokines and chemokines [103] and recruit monocytes from the periphery via TNFα-mediated microglia activation [104]. Acute intravenous injection of LPS (2.5 mg/kg) in mice induces a morphological transition from resting microglia into activated and round macrophage-like cells in the hypothalamus, thalamus and brainstem starting from ~8 h to 24 h after injection, with complete recovery at 7 days after injection [105]. In HFD obese mice, hypothalamic microglia undergo a biphasic activation showing a first inflammatory reaction at 2 weeks followed by a reduction of inflammatory markers, which return to higher levels after 4 weeks when bone marrow-derived myeloid cells gradually replace these cells [106]. In some circumstances, such as hepatic inflammation typical of severe obesity, peripheral circulating inflammatory cytokines like TNFα are required to stimulate microglia to produce MCP-1/CCL2 and cerebral monocytes [104,107] and to activate vagal afferent neurons to induce central inflammation [106]. HFD promotes microglia activation mainly in the Arcuate nucleus (ARC) of the hypothalamus by enhancing immunoreactivity to ionized calcium-binding adaptor molecule 1 (iba1) and cluster of differentiation 68 (CD68) and ramification of microglial processes. Of note, the microglia in the ARC of *db*/*db* and *ob*/*ob* obese mice are less prone to be activated, and leptin replacement rescues this condition in *ob*/*ob* mice pointing to a role of this adipokine in central obesity-driven inflammation. Intriguingly, HDF obese mice treated with glucagon-like peptide-1 receptor agonists show reduced microglial activation independently of body weight loss, indicating that diet-induced adipokines and gut signals affect the function of hypothalamic microglia in obesity [108]. Exposure to HFD has been associated with reactive gliosis (RG) and increased glial ensheathment of pro-opiomelanocortin (POMC) neuron perikarya [8]. This is of special relevance since astrocytes are tightly associated with pre- and post-synaptic sites forming a tripartite synapse [109,110], wherein they control and alter synaptic transmission by releasing gliotransmitters [68,111] acting at neuronal and/or glial receptors [112,113,114]. Accordingly, Horvath et al. [8] found quantitative and qualitative differences in the synaptology of POMC neurons between DR (diet-resistant) and DIO (diet-induced obese) animals, with a substantial reduction of excitatory inputs to POMC neurons in DIO and an enhancement in DR mice. The HFD-triggered disappearance of synapses on POMC neurons has been correlated with reactive glial ensheathment of the POMC perikarya, as revealed by electron microscopy [8]. In view of the above data, these results suggest that consumption of HFDs has a major repercussion on the cytoarchitecture of the hypothalamus in susceptible subjects, with changes that could lead to RG partly due to activation of the M1 phenotype of microglia. Indeed, in mice fed with a HFD, the number of reactive astrocytes and microglia in the ARC increased prior to weight gain [115]. Of note, diet composition is one of the main driving factors for RG in obesity as shown by recent studies. SCFAs were shown to reverse the effects of aberrant microglia activation [58], suggesting that the gut microbiota may govern centrally mediated events indirectly through regulation of monocyte trafficking to the brain and subsequent microglia activation, possibly via SCFA-mediated mechanisms. In mammals, leptin plays a two-fold role as a hormone and a cytokine. As a hormone, it modulates energy homeostasis by affecting energy expenditure, satiety and appetite [116,117,118]. As a pro-inflammatory adipokine, leptin is implicated, among others, in the modulation of gut microbiota composition through its receptors (LepRb) expressed in the submucosal intestinal zone, with increasing density of LepRb cells from the proximal tract to the colon [119]. TNFα and interleukin 6 (IL-6) stimulate the upregulation of leptin expression [120], which in turn can modulate production of cytokines from microglia [108,121] or directly target glial cells to facilitate reactive glial ensheathment and llial fibrillary acidic protein (GFAP) expression at the trypartite synapse [122]. Accordingly, the lack of leptin signaling in *ob*/*ob* and *db*/*db* mice caused deficiencies in several microglial functions in the hypothalamus, which could be recovered in *ob*/*ob* mice through leptin treatment. Interestingly, in both genetic models of obesity, the levels of most of the microglial activity markers are lower than in wild type mice, whereas IL1β is lower also in comparison with HFD obese mice. In fact, HFD mice express higher levels of IL1β in comparison with WT mice fed with chow diet [108]. The levels of TNFα are reduced in the hippocampus and in different areas of the brain of *db*/*db* mice [123]. SCFAs, produced by commensal dietary fiber-fermenting microbes characterizing healthy and lean individuals, induce the biosynthesis of leptin in adipocytes through the activation of GPR41 and subsequent leptin-mediated signaling [124]. Accordingly, GF mice show immature microglia that is unable to react to LPS activation [58], a condition that is reversed by SCFA treatment and is also lacking in GPR43 knockout mice. This finding suggests that microglia need constant stimulation from the gut microbiota to remain mature.

### 3.6. Enteroendocrine Signaling

Enteroendocrine cells (EECs) represent only 1% of epithelial cells in the GI tract, but are critical for the maintenance of gut homeostasis by regulating insulin secretion or food intake [125]. Among the EECs, the enteroendocrine L-cells and enterochromaffin cells are abundant in the distal small and large intestines in concomitance with the large amount of bacterial taxa localization. They can establish direct contact with the luminal constituents via the apical surface, including bacterial metabolites. In the postprandial state, enteroendocrine L-cells produce potent anorexigenic hormones, such as peptide YY (PYY) and glucagon-like peptide-1 (GLP-1), whose receptors are expressed locally in gut enteric neurons, vagal afferents, brain stem and hypothalamus [126,127]. The PYY and GLP-1 can reach the hypothalamus via the vagal–brainstem–hypothalamic pathway and then act directly or indirectly, where its specific receptors are expressed [128,129]. Recently, a basolateral cytoplasmic process of the L cells named “neuropod” has been characterized to form a synaptic contact with the enteric glia and vagal afferents. This synaptic connection between L cells and the enteric nervous system (ENS) suggests that signaling from the gut to the brain can act in a precise and fast way and that CNS may in turn regulate L cells. The postprandial spike of GLP-1 and PYY peptide release is activated by the luminal presence of carbohydrates via the sodium-coupled glucose transporter (SLC5A1) [130,131], long-chain fatty acids via free fatty acid receptors 1 (FFAR1) and 4 (FFAR4) [132,133] and monoacylglycerols via GPR119 [131] at the proximal luminal gut. On the contrary, in the distal gut, the activation of L cells is triggered almost exclusively by bacteria-derived metabolites. Moreover, SCFAs can stimulate the secretion of GLP-1 and PYY through FFAR2 (GPR43) and, to a lesser extent, the FFAR3 (GPR41) receptor [134,135]. Of note, obese individuals are generally characterized by decreased serum levels of both GLP-1 and PYY [126]. Changing gut microbiota through chronic dietary supplementation of prebiotics, mainly *Lactobacillus* spp. [136,137,138], or probiotics, mainly inulin as well as fructo-oligosaccharides (FOS) and galacto-oligosaccharides (GOS), was shown to increase the production of both GLP-1 and PYY [139,140] and to reduce body weight. Certain strains were capable of stimulating GLP-1 production both in vitro and in vivo.

### 3.7. Vagus Nerve

The vagal nerve (VN), a bundle of fibers afferent to (80%) and efferent from (20%) the brain, is regarded as the sixth sense because of its function in interoceptive awareness [141,142,143]. The VN is the principal component of the parasympathetic nervous system and it integrates peripheral information to the CNS through projections to the nucleus tractus solitarius (NTS), the first entrance of vagal afferents into the brain, as well as to the parabrachial nucleus, periventricular and lateral hypothalamus and amygdala, which are structures that form the central autonomic network (CAN) [143] (Figure 1). Electrical stimulation of vagal afferent fibers modifies brain concentration of serotonin, GABA and glutamate [144], causes a reduction of LPS-induced inflammation [145] and lowers body weight [146] similar to the activation of vagal fibers by SCFAs via GPR41 [147,148,149,150]. These effects suggest that probiotic-mediated VN activation could have a beneficial impact on neurotransmitter release and behavior. Of note, eCBs and eCB-like mediators are able to modulate the VN mechanosensitive response via cholecystokinin (CCK) and TRPV1 molecular pathways [151,152]. During obesity, high peripheral levels of eCBs inhibit the anorexigenic signal of the VN afferent fibers, and this effect is reversible since removal of the HFD for 12 weeks is able to reestablish the gastric vagal afferent mechanosensitivity [153]. Furthermore, in agreement with the anorexigenic role of leptin, this hormone potentiates the VN mechanosensitive response by promoting the sensation of satiety. An excess of leptin in mice made obese by a HFD can inhibit the tension-sensitive mechanoreceptors, thereby acting as an orexigenic signal [154]. LPS crosses the gut epithelium by reaching VN terminals expressing TLRs [155], whereas enteroendocrine cells, after stimulation with bacterial products [143], release serotonin, orexins (OX), CCK, glucagon-like peptide-1, peptide YY and ghrelin, all of which interact with vagal afferents by activating their receptors on the fibers [156]. A probable direct action of LPS on vagal afferent fibers via TLR4 at the level of the nodose ganglia has been hypothesized [157] and further evaluated in vitro and in vivo [143,158].

### 3.8. Enteric Nervous System

ENS is a network of neurons structured in the submucosal and myenteric plexus and is largely responsible for the coordination of gut motility [159]. ENS and CNS share structure and neurochemistry. They interact via intestinofugal projections to sympathetic ganglia via spinal and vagal routes afferent to the CNS [159]. These neural pathways represent the main gate for the delivery of gut microbiota-derived factors from the gut lumen to the CNS. Therefore, any mechanisms implicated in CNS dysfunction may also lead to ENS dysfunction or vice versa [160]. For instance, obesity or the presence of food in the gut inhibits vagal afferences to NTS via enteric nervous-mediated signaling from the gut to the brain [161], and regulates energy homeostasis by reducing c-fos expression in the NTS of obese rats. Given its function in signaling from microbiota to brain, it is highly likely that the vagus also plays a role in microbial regulation of food intake. A recent study highlighted the importance of microbiota-producing 5-HT in ENS neurogenesis [162] and how antibiotic treatment negatively impacts a wide range of structural and functional effects on neuronal and glial neurochemistry and function [163] of the ENS architecture.

## 4. Microbiota–Gut–Hypothalamic Axis

Alterations in food intake behaviors induced by obesity are associated with changes in gut microbiota composition, with increased abundance of Firmicutes and a decrease in Bacteroidetes [37,164,165,166]; similar changes in these phyla were observed with mice fed a high-fat and high-sucrose diet (HFHS) [167]. Healthy fats or polyunsaturated omega-3 and omega-6 fatty acids (PUFAs) induce beneficial metabolic effects by lowering the onset of cardiovascular diseases along with being protective against depression and cognitive decline [168]. The intake of PUFAs increases the healthy microbiota component (*Roseburia*, *Bifidobacteria* and *Lactobacillus* spp.) and prevents alterations of the gut microbiota profile post-antibiotic treatment [169]. On the other hand, gut microbiota can affect food intake behavior by producing proteins that mimic appetite-regulating peptides of the host, such as α-Melanocyte-stimulating hormone (α-MSH) [170,171]. *Escherichia coli* has been found to produce caseinolytic protease B (ClpB), which decreases short-term body weight and food intake associated with various psychopathological traits like anorexia nervosa, bulimia nervosa and binge-eating disorder [143]. Bariatric surgery has been shown to increase microbiota composition diversity [172] by enhancing the overall abundance of Gammaproteobacteria and Verrucomicrobia (to which the genus *Akkermansia* belongs), and reduction of Firmicutes in both human [172] and rodent [173] studies. Hypothalamic feeding circuits are highly sensitive to obesogenic diets. In particular, the mediobasal hypothalamus including the ARC is uniquely located close to the median eminence, a region of the brain with a highly fenestrated BBB through which microbe-released metabolites or microbe-induced signals under obesogenic diets are more easily sensed by neurons. As a consequence, high-fat and high-carbohydrate diets stimulate orexigenic neuropeptide Y (NPY)/agouti-related peptide (AgRP) neurons to produce advanced glycation end products, which activate TNFα and related microglia reactivity, with dysfunction of anorexigenic neurons [173]. Conversely, weight loss, anorexia and sickness behavior are triggered by TLR2-mediated activation of hypothalamic microglia, which reduce GABAergic inputs to anorexigenic POMC neurons [174]. Our group found an obesity-related rewiring of orexigenic orexin/hypocretin (OX) neurons in the LH as result of a switch from predominantly excitatory to inhibitory inputs in obese mice [175]. As mentioned above, in their role of brain’s surveillance, microglial cells dynamically elongate and retract their branches, also contacting synapses [176]. Variations in the reactivity and/or distribution of hypothalamic astrocytes also affect synaptic organization and POMC responsiveness to glucose [177,178] or leptin [179,180] (Figure 2). These changes might also affect eCB and OX levels and subsequent responses of POMC to metabolic hormones [179,181]. Understanding how alterations in the immune status induced by dysbiosis may support a vicious “obesity ↔ neuroinflammation” circle in the hypothalamus will provide novel clues to develop probiotic and prebiotic interventions functioning as novel treatments for obesity by restoring gut microbiota composition and gut–brain signaling [182].

## 5. Microbiota–Endocannabinoidome–Gut Axis

The gut microbiota interacts with the host through different biochemical mechanisms. The eCB(ome) or “expanded eCB system” is a pleiotropic endogenous system of signaling lipids composed of numerous long-chain fatty acid-derived mediators with chemical similarity to eCBs and their receptors and metabolic enzymes, which are often shared with eCBs [183]. It has important physiological roles in the modulation of gastrointestinal function, energy metabolism, behavior and, as it is being increasingly suggested, host cell–gut microbe communications. Treatment with prebiotics and probiotics can affect host eCB levels [184], whereas as pointed out above, some commensal bacteria produce eCB-like metabolites functioning at the same receptors as their host cell counterparts and could be considered as part of the eCBome [101]. GF mice, which lack most of their microbiome, differ drastically from conventionally raised (CR) mice in the expression levels of several eCBome receptors and enzyme mRNAs and in intestinal concentrations of eCBome mediators. Importantly, most of these alterations are reversed after only one week by fecal microbiome transfer from CR mice, strongly suggesting that commensal microorganisms directly control intestinal (and particularly small intestinal) eCBome signaling. This interaction seems to particularly impact CB1, PPARα and GPR55 signaling, CB1 and PPARα signaling is increased in the small intestine, whereas GPR55 signaling is decreased in GF mice [28]. On the other hand, genetic or pharmacological inactivation of eCBome mediator biosynthesis or action influences gut microbial composition and the host’s metabolic response to high-fat diets [29,185,186,187]. The eCBome is important in the regulation of gut permeability since blocking CB1 activity in mice fed with obesogenic diet inhibited the development of obesity, improved glucose homeostasis and ameliorated intestinal permeability by lowering circulating LPS levels and inflammatory cytokine profile in parallel with enhanced *Akkermansia muciniphila* and decreased *Lachnospiraceae* in the gut [29]. Muccioli et al. showed that CB1 agonism in wild-type mice increased gut permeability, whereas CB1 antagonism partially rescued tight junction integrity within the intestinal epithelium and reduced plasma LPS levels [1]. Other eCBome receptors, such as TRPV1 and PPARα, instead ameliorate intestinal barrier integrity [188]. Therefore, a diet–microbiota–eCBome loop influences gut permeability and systemic inflammation in opposite ways, by creating either (i) a vicious circle, following obesogenic diets that enhance endotoxemia and metabolic diseases [189] or (ii) a virtuous circle, following the consumption of prebiotic fiber-associated diets, which reduces intestinal permeability, metabolic endotoxemia and systemic inflammation. Indeed, several studies have suggested a close relationship between LPS and the eCB system. CB1 receptors are widely distributed in the gut wall [190] and the levels of their main ligands, AEA and 2-AG, can be modulated by LPS, since in vitro treatment with ionomycin (5 µM) and LPS (200 µg/mL) cause respectively a 24-fold and 2.5-fold enhancement of 2-AG levels in J774 macrophages [191]. In human lymphocytes, LPS enhances AEA levels by 4-fold in comparison with vehicle-treated cells [17], mainly by downregulation of FAAH gene expression at the transcriptional level [18]. Although CB1 receptor control of gut permeability seems to depend on the portion of gut wherein they are expressed [192], changes in eCB tone that restore normal gut permeability have been first described in the whole intestine of GF mice or obese mice after prebiotic ingestion [193,194,195]. Similar results were obtained by blocking the CB_1_ receptor in obese mice, which ameliorated gut barrier function by improving the expression of tight junction proteins (ZO-1 and occluding) [196,197,198]. Since obesity is commonly characterized by increased eCB tone, higher plasma LPS levels, altered gut microbiota and metabolic dysfunction of the adipose tissue, it is likely that all these changes could be primarily triggered by an alteration of the eCB system [46,58,173]. Accordingly, Mehrpouya-Bahrami and collaborators [29] demonstrated that CB1 antagonism in mice made obese by HFD counteracts dysbiosis and reduces LPS levels, thereby helping to counteract the obese phenotype. Some controversy regarding the role of LPS in obesity arose when Dalby and colleagues did not detect any protection against increased body weight in studies performed in TLR4 knockout mice, which were unresponsive to LPS signaling [199]. These contradictory results could be due to differences in diet, baseline microbiota composition or genetic background. Importantly, the CB1 agonist HU‑210 increased both adipogenesis and circulating levels of LPS, with final enhancement of gut permeability. On the contrary, rimonabant, by reducing LPS plasma levels and inflammation, improved gut barrier permeability in *ob*/*ob* mice [29,118]. More recently, time- and gut segment-specific microbiome disturbances concomitant to modifications of intestinal and circulating eCBome mediators have been found after 56 days of HFHS diet, thus suggesting the involvement of the microbiota–gut–eCB(ome) axis in diet-induced glucose intolerance, obesity and other metabolic disturbances. Elevation of AEA, in both ileum and plasma, and 2-AG, in plasma, as well as alterations in several other NAEs and 2-acylglycerols were reported by comparison with low-fat, low-sucrose (LFLS) diet with identical fiber and fatty acid percent compositions [27]. In the ileum, HFHS diet-induced temporal changes in AEA levels, and hence possibly in CB1 tone, inversely correlated with the abundance of metabolically beneficial gut commensal microorganisms, such as *Akkermanisa*, *Barnesiella*, *Eubacterium*, *Adlercreutzia*, and *Propionibacterium* spp., in some cases independently of body weight changes. This study revealed the existence of interactions between the eCBome and several intestinal bacterial species during early and late HFHS-induced dysmetabolic events, with potential impact on the capability a host to adapt to increased intake of fat and sucrose.

## 6. Conclusions

The microbiota–eCB(ome)–gut–brain axis is a complex communication system mediated by lipid, hormonal, pro-/anti-inflammatory and neural signals. Mounting evidence is revealing the effects of diet composition on the gut microbiota and microbiota-derived metabolites with consequent impact on brain function. When the gut bacteria are exposed to obesogenic diets, alterations of gut microbial composition or dysbiosis occurs, which impairs the integrity of intestinal barrier (“leaky gut”) with consequent facilitation of LPS translocation through the gut barrier into the lumen and rise of circulating pro-inflammatory cytokines or endotoxemia [7,200]. Pro-inflammatory cytokines (e.g., IL1β and TNFα) and neurotrophic factors, starting from the mesenteric lymph nodes, contribute to BBB breakdown, thus favoring the infiltration of leukocytes into the CNS and promoting the development of neuroinflammation. Moreover, HFD-induced adipogenic responses, by causing hypertrophy of the adipose tissue, boost a pro-inflammatory chronic response, which constitutes an important link between obesity and pathophysiological sequelae [201]. Leaky gut and ensuing endotoxemia and adipose tissue-related inflammation may generate hyperglycemia and insulin resistance, which are essential to initiate a sequence of pathophysiological events in obesity [201]. Diets poor in fibers or vitamins and rich in high-calorie nutrients, such as the Western diet, together with lack of physical activity, will negatively regulate energy metabolism and, hence, contribute to the development of metabolic syndrome [202], including obesity, visceral adipose tissue deposition [203], glucose intolerance, pre-diabetes and type-2 diabetes [204], dyslipidemia [205], hypertension [206], atherogenic inflammation [207] and cardiovascular disorders [208] up to the onset of cognitive impairment [209] and behavioral disorders. Results from many different laboratories, together with a plethora of epidemiological data, point to the control of the diet–microbiota–eCB(ome)–gut–brain axis as a major predictor of healthy metabolism. Hence, fighting bad dietary and lifestyle habits and introducing dietary fiber and vitamin supplements may lead to the counteraction of many signs of metabolic syndrome [210,211,212].

## Figures and Tables

**Figure 1 ijms-21-01554-f001:**
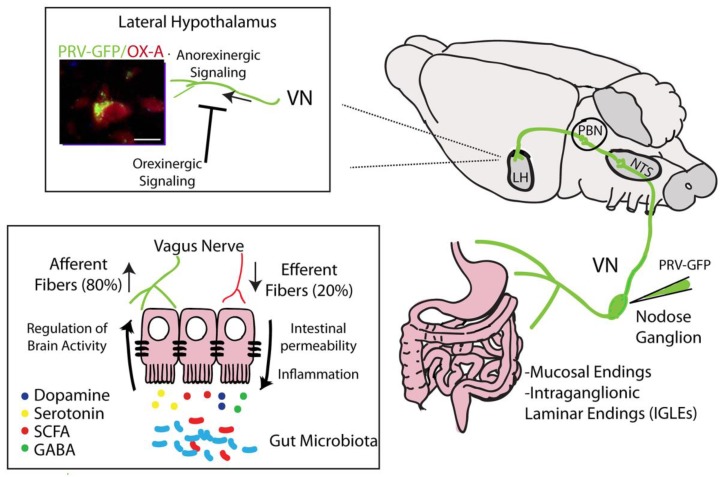
Identification of projections from the nodose ganglion to hypothalamic neurons (orexin/hypocretin neurons) by injection of pseudorabies virus (PRV)-GFP. Schematic representation of hypothalamic circuits, wherein the vagal nerve (VN) drives anorexinergic signaling. These circuits are, in turn, inhibited by hypothalamic orexigenic molecules (i.e., endocannabinoids). Right panel: Vagal afferences coming from the myenteric plexus and mucosal endings projecting to the nodose ganglia. At the supraspinal level, PRV-GFP labeling has been found in the nucleus tractus solitarius (NTS), parabranchial nucleous (PBN) and lateral hypothalamus (LH). Left panel: Co-localization of PRV-GFP labeling with Orexin-A (OX-A) immunoreactivity in the lateral hypothalamus (scale bar: 15 mm). Vagal efferent fibers regulate gut permeability and inflammation influencing gut functions; conversely, the gut microbiota affects brain activity via VN afferent fibers through the synthesis and secretion of molecules such as dopamine, serotonin, short-chain fatty acids (SCFAs) and γ-aminobutyric acid (GABA).

**Figure 2 ijms-21-01554-f002:**
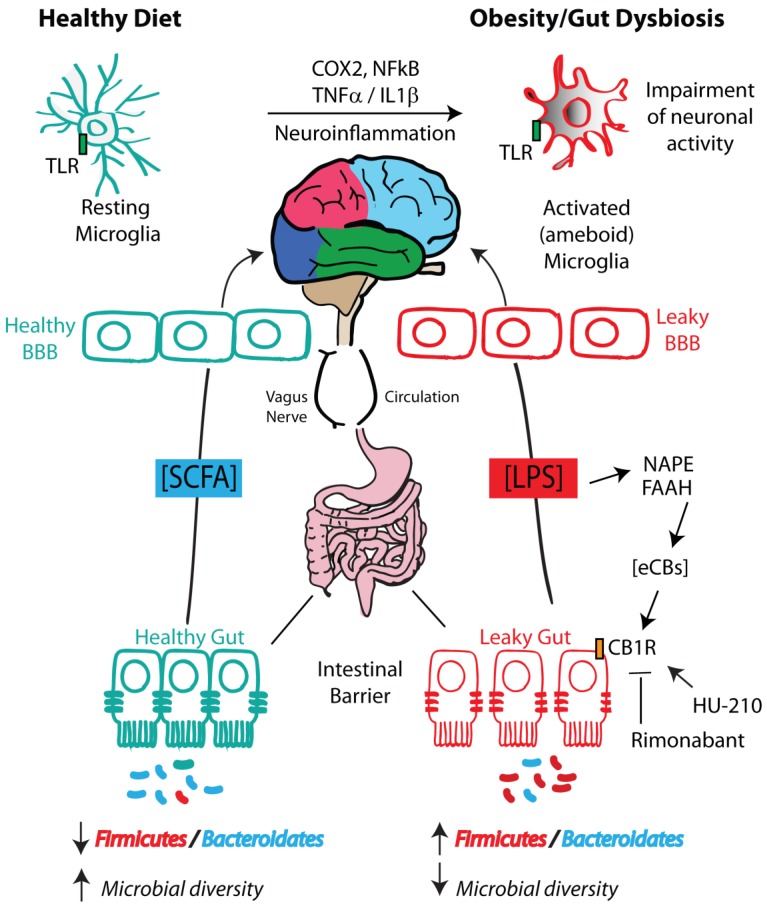
Diet–microbiota–eCBome–gut–hypothalamic axis. The diet influences gut microbiota composition and regulates intestinal permeability. Obesogenic diets increase the Firmicutes/Bacteroidetes ratio, and cause lipopolysaccharide (LPS) production by raising the Gram-negative bacterial component. Cannabinoid receptor 1 (CB1) overactivation by endocannabinoids (eCBs), such as the N-acyl-phosphatidylethanolamine (NAPE)-derived anandamide (AEA), may further enhance high-fat diet (HFD)-induced enhancement of gut permeability, although other NAPE-derived *N*-acylethanolamines (NAEs) may oppose this effect by acting at transient receptor potential channel subfamily V member 1 (TRPV1) and peroxisome proliferator-activated receptor α (PPARα) (not shown). Hypothalamic microglia undergo activation, thereby showing enhanced immunoreactivity to ionized calcium-binding adaptor molecule 1 (iba1) and cluster of differentiation 68 (CD68) and ramification of microglial processes. Peripheral circulating inflammatory cytokines, such as tumor necrosis factor α (TNFα), stimulate microglia to produce MCP-1/CCL2 chemokine and cerebral monocytes to further activate vagal afferent neurons and induce central inflammation. SCFAs, produced following correction of the HFD with fiber-derived diet, ameliorate these dysfunctions through effects on the blood–brain barrier (BBB) and microglia.

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
