# Peer review of "Obesity Affects the Microbiota–Gut–Brain Axis and the Regulation Thereof by Endocannabinoids and Related Mediators"

_ijms, 2020, doi:10.3390/ijms21051554_

Round 1

Reviewer 1 Report

The manuscript entitled "Obesity affects the microbiota-gut-brain axis and the regulation thereof by endocannabinoids and related mediators" is a complete, comprehensive review of the most recent advancements in the kwoledge of the enteric microbiota-nervous system connection. Particularly noteworthy are the paragraphs around the enteroendocrine system and the vagus nerve. The article is worth publication in its present form. 

Author Response

Dear Reviewer,

thank you for all your comments and good considerations regarding the manuscript.

Best regards

Luigia Cristino

Reviewer 2 Report

The manuscript entitled “Obesity affects the microbiota-gut-brain axis and the regulation thereof by endocannabinoids and related mediators” reviews about microbiota-gut-brain axis in obesity, with the respect to molecules, involved in this axis.

This paper is organized and written well. However, there are many shortcomings and grammatical mistakes. The words amino acids are written separately. Please, correct throughout the manuscript. In vitro and in vivo should be written in Italics. For example, line 38:..bulk of research is this field …(is instead of in), line 82: ..the present review, aims… without comma, line 37: acetate [37] . After the parenthesis should immediately follow dot. This mistake occurs many times in the text. Please check. There are some places in the manuscript, where more citations are needed, because authors refer it. For example, lines 121-125 refer that mechanistic studies, but they cite only one reference. 258-260 the authors have no reference at all. Please add. On the other hand, in the lines 130-131 the authors wrote: a study reported and cite two. Please correct. In the lines 148-149, a word is missing (Since then, it known). In the text, there have to be all abbreviations explained in the same format. In the line 162, the abbreviation BBB is not in the parenthesis at all. CNS is used only in some places of the manuscript, otherwise is written as central nervous system. Please, correct it. In the line 302, there are abbreviations used as first, thereafter in the parenthesis, there was the explanation. The format used in the other parts was different. Some abbreviations are not explained at all, f. e. FFAR1 and FFAR4 receptors and many others. Cholecystokynin is once used as abbreviation, once as the full word.

The word broth in the line 262 means soup. I am definitely sure that the word should not stay in the text. Please check it. In the line, the word  therenyu is not known to me.  In the line 436, the sentence should begin with the capital letter (our group).

Figure 2 – in the text, there is a part what cannot be copied or marked at all and is written with other font. Please correct it.

Author Response

Dear Review,

Thank you for all your comments and considerations.

We made all the corrections you requested for the manuscript. Furthermore, other grammatical errors have been revised throughout the text.

We hope that all the changes have been accomplished satisfactorily.

Please see the attachment of the revised text.

Best regards

Luigia Cristino